# History of heart failure and chronic kidney disease and risk of all-cause death after COVID-19 during the first three waves of the pandemic in comparison with influenza outbreaks in Sweden: a registry-based, retrospective, case–control study

Viveca Ritsinger [1,2] Johan Bodegård,[3] Robin Kristofi,[4] Marcus Thuresson,[5] David Nathanson,[6] Thomas Nyström,[7] Jan Eriksson,[4] Anna Norhammar[1,8]

**Correspondence to**
Dr Viveca Ritsinger;
viveca.ritsinger@ki.se

## ABSTRACT

**Objectives** To explore how cardiorenal disease (CRD; heart failure and/or chronic kidney disease) impacted mortality in men and women hospitalised for COVID-19 during the first three waves of the pandemic in Sweden in comparison to previous influenza outbreaks.

**Design** A registry-based, retrospective, case–control study.

**Setting** Hospital care in Sweden.

**Participants** All patients in Sweden with a main hospital diagnosis of COVID-19 (January 2020–September 2021) or influenza (January 2015–December 2019) with previous CRD were identified in registries and compared with a reference group free from CRD but with COVID-19 or influenza.

**Primary outcome measure** Associated risk of all-cause death during the first year was analysed using adjusted Cox proportional hazards models.

**Results** In COVID-19 patients with and without prior history of CRD (n=44 866), mean age was 79.8 years (SD 11.8) and 43% were women. In influenza patients (n=8897), mean age was 80.6 years (SD 11.5) and 45% were women. COVID-19 versus influenza was associated with higher mortality risk during the first two COVID-19 waves (HR 1.53; 95% CI 1.45 to 1.62, p<0.001 and HR 1.52; 95% CI 1.44 to 1.61, p<0.001), but not in the third wave (HR 1.07; 95% CI 0.99 to 1.14, p=0.072). CRD was an independent risk factor for all-cause death after COVID-19 in men and women (men: 1.37; 95% CI 1.31 to 1.44, p<0.001; women: 1.46; 95% CI 1.38 to 1.54, p<0.001). At ages <70 years, women with CRD had a similar mortality rate to men with CRD, while at ages ≥70 years, the mortality rate was higher in men.

**Conclusions** Outcome after COVID-19 is worse if CRD is present. In women at ages <70 years, the presence of CRD attenuates the protective effect of female sex. COVID-19 was associated with higher mortality risk than influenza during the first two pandemic waves.

## STRENGTHS AND LIMITATIONS OF THIS STUDY

⇒ This study includes all patients with cardiorenal disease hospitalised for COVID-19 during the first three pandemic waves in Sweden.

⇒ This study compares mortality associated with COVID-19 to previous influenza outbreaks.

⇒ Sex-based analyses by age group were performed to explore the importance of sex as a risk factor.

⇒ This was a national registry-based study that lacked data regarding the proportion of patients vaccinated for influenza.

⇒ The study also lacked information on laboratory analyses and variables such as body mass index and type of heart failure.

## INTRODUCTION

Infection with SARS-CoV-2 (COVID-19) globally affects people differently with some individuals more severely affected in need of hospitalisation and in more unfortunate circumstances, causing early mortality and reduced life expectancy. Starting rapidly during the first months after the initial reports from the outbreak in China,[1 2] and thereafter, there are multiple reports on COVID-19 disease course and consequences, treatment aspects, pathophysiology and characteristics of high-risk individuals. Early on it was reported that risk factors associated with hospitalisation and mortality from COVID-19 were characteristics such as advanced age, male sex, hypertension, cardiovascular disease, chronic kidney disease (CKD) and obesity.[3–7] Recently, asthma and Down's syndrome were added to this list,[8] while

diabetes has been reported as a risk factor in some studies, while other studies found this risk to be attributed to other comorbidities.[9–12] Due to the fact that coexistence of risk factors and multiple diseases often cluster at more advances ages, earlier reports due to low numbers and limited information, could not clarify independent associations. Some reports have been further limited to data on patients with COVID-19 admitted to intensive care, thus possibly introducing selection bias due to aspects of national and local resources and recommendations for such care, including only individuals judged to be in favour of intensive care.[7] Therefore, there is a need of more extensive and deeper analysis on contributing risk factors for severe COVID-19 outcomes in a less selected cohort to enhance present knowledge on important risk factors. Such information is of importance for improving future preventive measures. In this study, we aimed to explore how history of heart failure and CKD (ie, diseases that are notably inter-related and with high risk of cardiovascular and all-cause death[13]), impacts the course of COVID-19 infection and subsequent mortality during the first three waves of COVID-19 in Sweden, the only western country that did not apply a lockdown during the first pandemic year. Further, we aimed to explore this risk in comparison to previous influenza outbreaks and with a sex perspective.

## METHODS
### Study design
A registry-based, retrospective, case–control study.

### Data sources
Sweden has a comprehensive, nationwide public healthcare system. All citizens have a unique personal identification number (person-ID), which is mandatory for all administrative purposes (including any contact with the healthcare system and drug dispensaries), thus providing a complete medical history from a population perspective. This study included data from Statistics Sweden, National Prescribed Drug Register, the National Cause of Death Register and the National Patient Register covering all hospitalisations with discharge diagnoses and all outpatient hospital visits. Individual patient-level data from the national registers were linked using the person-ID by the National Board of Health and Welfare. The linked anonymised database was managed separately by Statisticon AB, Uppsala, Sweden.

### COVID-19 and influenza population
The study population includes all patients with cardiorenal disease (CRD) from the above specified CARE-19 cohort with a main inpatient hospital diagnosis of COVID-19 (1 January 2020–9 September 2021) or influenza (1 January 2015–31 December 2019). Thus, the study population includes all patients with CRD hospitalised for COVID-19 in Sweden and around two-thirds of the total hospitalisations for COVID-19. Of note, the reference group is not chosen based on the COVID-19 or influenza date, but on the CRD date, as described below.

### Study population
All patients from the CARE-19 cohort above 18 years of age with a main inpatient hospital diagnosis of COVID-19 (1 January 2020–9 September 2021) or influenza (1 January 2015–31 December 2019) were identified in the National Patient Register.

COVID-19 and influenza hospitalised patients were identified with any of the following codes as main diagnosis: B34.2, B97.2, J09.9, J10–J11, J20.8, U04.9, U07.1, U07.2 or ZV100 (ZV100 is a code used in Sweden in particularly during the initial waves of the pandemic for COVID-19 and later on more for procedures related to COVID-19). Furthermore, patients are only included once in each group: at the first COVID-19 or influenza hospitalisation. However, a prior influenza hospitalisation did not exclude the patient if later hospitalised with COVID-19.

### CARE-19 cohort database
This database includes all patients hospitalised with heart failure and/or CKD as a main diagnosis recorded in national registries in Sweden since 1995 to 9 September 2021 that also were alive on 1 January 2010. For these patients, a reference group without CRD was collected. This reference group was matched by age and sex with cases at the time point for CRD if diagnosed between the years 2010–2020, if diagnosed before 2010 then 1 January 2010 was used as time point for collecting references. Patients with heart failure were defined having at least one hospital diagnosis of heart failure: ICD-10 code I50, I11.0, I13.0, I13.2 and patients with CKD were defined as having at least one hospital diagnosis of CKD: N17–N19, I12.0–I12.9, I13.1, I13.2, N08.3, E10.2, E11.2, E12.2, E13.2, E14.2, Z49, Z99.2.

### Study period
Index date was defined as the date of COVID-19 or influenza hospital admission, which is the reference date for all analyses. Baseline comorbidities were searched based on all hospital data from 1997 when ICD-10 was introduced until the day prior to index, and baseline medications were searched during 1 year prior to index date until the day prior to index.

Follow-up for events started at the index date (including) and ended at 1 year, or at date of death or end of follow-up defined as 31 December 2019 for influenza patients and 9 September 2021 for COVID-19 patients.

### Statistical methods
Description of baseline characteristics was performed using mean and SD for numerical variables, except for duration of first hospital stay which is described as median and IQR. Categorical variables are presented as number and per cent. Baseline tables are stratified by type of diagnosis (COVID-19 or influenza) and CRD status (CRD/without CRD). The cumulative risk of all-cause death

is presented in Kaplan-Meier curves stratified by type of diagnosis (COVID-19 or influenza) and CRD status (CRD/without CRD).

The relative risk of all-cause death was analysed using a multivariable Cox regression model where the model in addition to COVID-19 versus influenza and CRD versus no CRD also included the following variables; age, sex, use of renin–angiotensin system inhibitors (RAAS inhibitors), coronary artery disease, stroke, peripheral artery disease (PAD), diabetes, pneumonia, chronic obstructive pulmonary disease (COPD), vitamin D-deficiency, corticosteroids, paracetamol and diagnosed obesity.

To explore the importance of risk factors relative to sex, separate analyses were in addition performed within men and women and by age group.

An exploratory analysis was performed to evaluate if the risk of all-cause death has changed during the cause of the COVID-19 pandemic period where the COVID-19 inclusion period was split into three-time intervals separated when the frequency of hospitalisations was low. Thus, the time intervals were intended to capture the first three waves of the pandemic in Sweden. The time intervals used: wave 1: 1 January 2020–31 August 2020; wave 2: 1 September 2020–31 January 2021 and wave 3: 1 February 2021–9 September 2021. To limit complexity, the analysis was only performed within the CRD diagnosed patients. The cumulative incidence of all-cause death by wave is presented in Kaplan-Meier curves with Influenza as reference. In addition, as the ICD-10 code ZV100 was only used in Sweden a sensitivity analysis was performed excluding patients with a main diagnosis of ZV100.

All output from the statistical models is presented with 95% CIs and p values. No adjustment for multiplicity has been performed.

### Patient and public involvement
None.

## RESULTS
### Characteristics of COVID-19 patients with heart failure and/or CKD
In all, 44 866 patients hospitalised with a main diagnosis of COVID-19 were identified. Of those, 23 649 had a prior history of heart failure, 11 313 had a prior history of CKD, 5767 had a prior history of the combination of heart failure and CKD and 15 671 references were free from previous CRD. Baseline characteristics for patients are presented in table 1.

Patients with COVID-19 and CRD also had other comorbidities as atrial fibrillation 53%, coronary artery disease 49%, cancer 38%, diabetes 34%, previous pneumonia 38%, stroke 28%, COPD 21%, vitamin D deficiency 20%, history of thromboembolism 14% and PAD 12%. Cardiovascular treatment prior to hospitalisation was commonly used with heart failure drug treatment in 90% (any of ACE inhibitors, angiotensin II receptor blockers,

mineralocorticoid receptor antagonists, angiotensin receptor-neprilysin inhibitors, sodium-glucose cotransporter-2 inhibitors and beta blockers). Statins were used in 52% and corticosteroids in 29%.

### Characteristics of COVID-19 patients without heart failure and/or CKD (reference group)
Conversely, in the reference group (COVID-19 without any history of heart failure or CKD) mean age was 81.2 years (SD 11.3) and 45% were women. Except for cancer the burden of other comorbidities was less compared with COVID-19 patients with CRD.

### Characteristics of patients with influenza with heart failure and/or CKD
N=5983 patients with a history of CRD were hospitalised for influenza. Their mean age was 79.5 (SD 12.3) years and 45% were women. Apart from a phistory of pneumonia, reported in 47% in influenza patients with CRD compared with 38% in COVID-19 patients without CRD, patients hospitalised for influenza and CRD the years before the COVID-19 pandemic had a very similar comorbidity pattern as COVID-19 patients with CRD.

### Outcome
During a mean follow-up period of 6.9 months (SD 4.3), 16% (n=7085) of the patients with COVID-19 died. All-cause death rate (death per 100 person years) for patients with COVID-19 and CRD was 245 compared with 196 in references without CRD. When excluding patients with a COVID-19 main diagnosis of ZV100 (n=32 995) all-cause death rate (death per 100 person years) for patients with COVID-19 and CRD was even higher; 537 compared with 375 in references without CRD (mean follow-up time 6.1 (SD 4.5) months).

Corresponding mortality rate for influenza between the years 2015 and 2019 was lower (119 in those with CRD vs 104 in those without).

Details on cumulative event rates for all-cause death in patients with COVID-19 or influenza stratified by previous CRD are presented in figure 1. A higher death rate was seen for patients with CRD compared with references without with the highest rate in patients with CRD with COVID-19. Except for CRD (1.41; 95% CI 1.36 to 1.46, p<0.001), other risk factors associated with all-cause death in patients with COVID-19 were age (1.61; 95% CI 1.58 to 1.64, p<0.001), previous pneumonia (1.19; 95% CI 1.16 to 1.23, p<0.001) and diabetes (1.13; 95% CI 1.09 to 1.17, p<0.001; online supplemental table 1). Female sex was associated with a reduced risk of all-cause death (0.76; 95% CI 0.73 to 0.78, p<0.001) while no association was seen for coronary artery disease (1.00; 95% CI 0.97 to 1.03, p=0.881). In an adjusted Cox regression analysis in the total study cohort COVID-19 versus influenza was associated with an increased risk of all-cause death (HR 1.54; 95% CI 1.47 to 1.61, p<0.001). When excluding patients with COVID-19 with a main diagnosis of ZV100 corresponding HR (95% CI)

**Table 1** Baseline characteristics in patients with COVID-19 or influenza with and without cardiorenal disease (CRD)

| | COVID-19 with CRD n=29 195 | COVID-19 without CRD n=15 671 | Influenza with CRD n=5983 | Influenza without CRD n=2914 |
|---|---|---|---|---|
| Age, years, mean (SD) | 79.0 (11.9) | 81.2 (11.3) | 79.5 (12.3) | 83.1 (9.4) |
| Age, years, median (SD) | 81 (18–107) | 83 (18–106) | 82 (20–106) | 84 (24–107) |
| Females, n (%) | 12 474 (43) | 7001 (45) | 2700 (45) | 1276 (44) |
| Year | | | | |
| 2020, n (%) | 19 729 (68) | 9914 (63) | 0 (0) | 0 (0) |
| 2021, n (%) | 9466 (32) | 5757 (37) | 0 (0) | 0 (0) |
| 2015, n (%) | 0 (0) | 0 (0) | 946 (16) | 443 (15) |
| 2016, n (%) | 0 (0) | 0 (0) | 861 (14) | 374 (13) |
| 2017, n (%) | 0 (0) | 0 (0) | 1154 (19) | 604 (21) |
| 2018, n (%) | 0 (0) | 0 (0) | 1890 (32) | 963 (33) |
| 2019, n (%) | 0 (0) | 0 (0) | 1132 (19) | 530 (18) |
| In-hospital stay, days, median (IQR) | 6 (4–11) | 6 (4–10) | 6 (4–10) | 6 (4–10) |
| Heart failure, n (%) | 23 649 (81) | 0 (0) | 4971 (83) | 0 (0) |
| Chronic kidney disease, n (%) | 11 313 (39) | 0 (0) | 2026 (34) | 0 (0) |
| Dialysis, n (%) | 1777 (6) | 0 (0) | 520 (9) | 0 (0) |
| Coronary artery disease, n (%) | 14 314 (49) | 3605 (23) | 3193 (53) | 786 (27) |
| Stroke, n (%) | 8149 (28) | 3655 (23) | 1691 (28) | 732 (25) |
| Atrial fibrillation, n (%) | 15 517 (53) | 3178 (20) | 3177 (53) | 645 (22) |
| Peripheral artery disease, n (%) | 3567 (12) | 819 (5) | 681 (11) | 158 (5) |
| Diabetes, n (%) | 10 050 (34) | 2974 (19) | 1843 (31) | 580 (20) |
| Chronic obstructive pulmonary disease, n (%) | 6082 (21) | 1421 (9) | 1337 (22) | 324 (11) |
| Any history of pneumonia (viral/bacterial), n (%) | 11 075 (38) | 2937 (19) | 2789 (47) | 855 (29) |
| Any history of thromboembolism (DVT or PE), n (%) | 3944 (14) | 1407 (9) | 792 (13) | 270 (9) |
| Vitamin-D deficiency, n (%) | 5780 (20) | 1114 (7) | 1084 (18) | 156 (5) |
| Cancer, n (%) | 10 949 (38) | 5780 (37) | 2081 (35) | 1014 (35) |
| Any of ACEi/ARB/MRA/ARNi/SGLT2i/β-blocker, n (%)* | 26 187 (90) | 9677 (62) | 5401 (90) | 1863 (64) |
| RAAS inhibitor, n (%) | 19 285 (66) | 7122 (45) | 4073 (68) | 1299 (45) |
| Statins, n (%) | 15 170 (52) | 5566 (36) | 2934 (49) | 1028 (35) |
| Corticosteroids, n (%) | 8512 (29) | 2910 (19) | 2096 (35) | 699 (24) |
| Paracetamol, n (%) | 17 316 (59) | 7325 (47) | 3495 (58) | 1384 (47) |
| Modulating antineoplastic and hormone drugs, n (%) | 1698 (6) | 962 (6) | 299 (5) | 145 (5) |

*Any of ACE inhibitors, ARBs, MRA, ARNi, SGLT2 inhibitors and beta blockers.
ARBs, angiotensin receptor blockers; ARNi, angiotensin receptor-neprilysin inhibitors; DVT, deep vein thrombosis; MRA, mineralocorticoid receptor antagonists; PE, pulmonary embolism; RAAS, renin–angiotensin system inhibitors; SGLT2, sodium-glucose cotransporter-2.

for COVID-19 versus influenza was 2.23 (2.12 to 2.34, p<0.001) while other risk factors were similar (online supplemental table 2).

Figure 2 depicts cumulative incidence of all-cause death within patients with CRD by the three COVID-19 waves where a similar event rate was seen in the first two waves but with a significantly lower death rate in the third wave although still higher than that observed for influenza. Compared with influenza, COVID-19 was associated with an increased mortality risk in the first two waves (1.53; 95% CI 1.45 to 1.62, p<0.001 and 1.52; 95% CI 1.44 to 1.61, p<0.001) but not in the third wave (1.07; 95% CI 0.99 to 1.14, p=0.072).

## Sex-specific analyses in patients with COVID-19 and influenza with CRD

Baseline characteristics for men and women are depicted in online supplemental tables 3 and 4. Mean age in men was 77.7 years (SD 11.7) and 80.9 years (SD 12.0) in women with even higher mean age in references without previous CRD (men 80.1 (SD 10.4) and women 82.6 years (SD 12.3)). Mean age in influenza patients with CRD was 78.1 (SD 11.9) years in men and 81.1 (SD 12.4) years in women.

Comorbidities in COVID-19 and influenza were equally distributed among men and women apart from coronary artery disease which was more common among men with

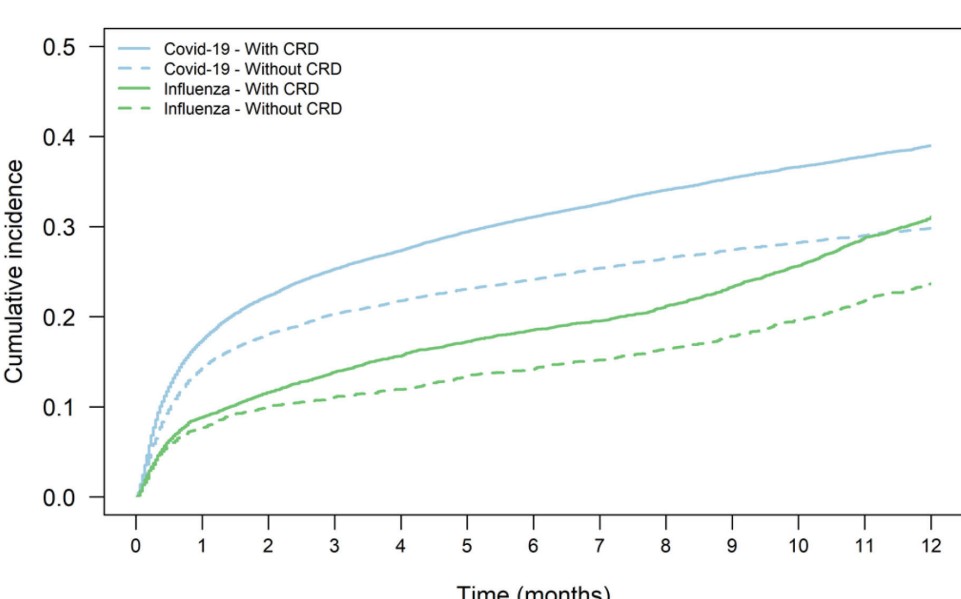

**Figure 1** Cumulative incidence of all-cause death in COVID-19 patients between the years 2020 and 2021 with cardiorenal disease (CRD; blue) and without CRD (dotted blue) and in influenza patients between the years 2015 and 2019 with CRD (green) and without CRD (dotted green).

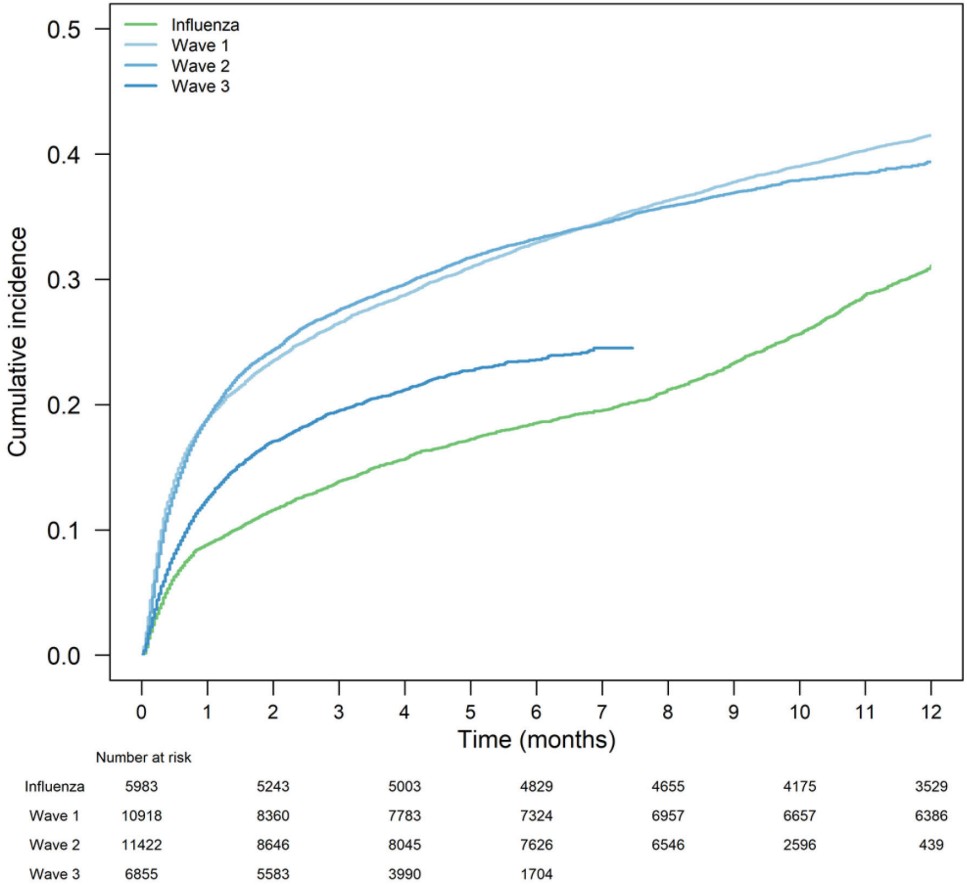

**Figure 2** All-cause death in patients with cardiorenal disease and COVID-19 or influenza by the three COVID-19 waves in Sweden.

A

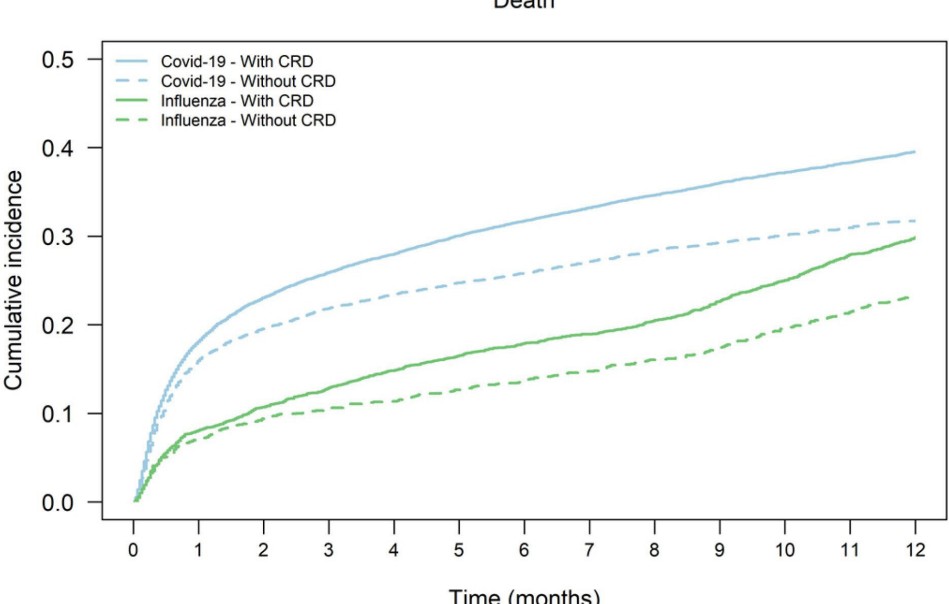

B

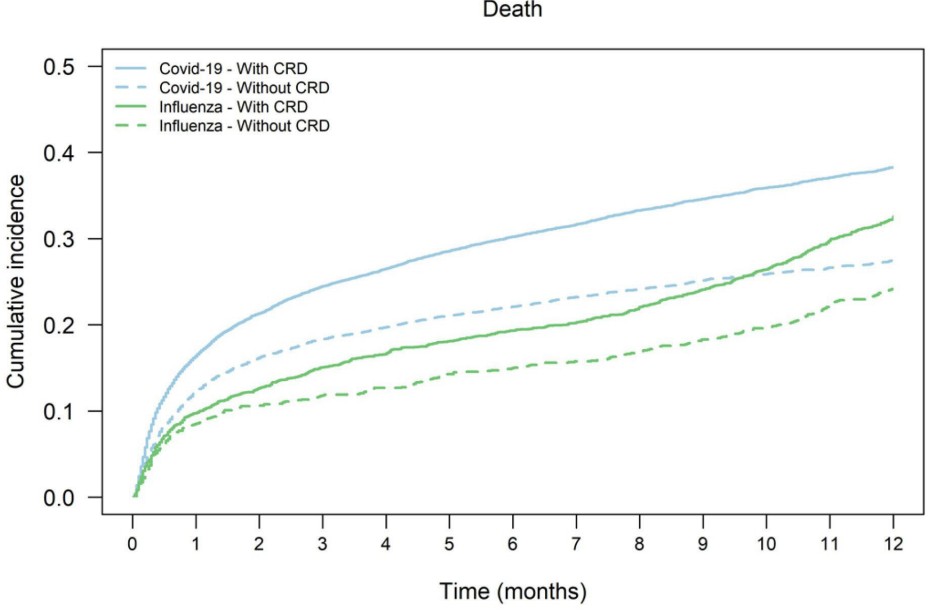

**Figure 3** Cumulative incidence of all-cause death in men (A) and women (B) with COVID-19 between the years 2020 and 2021 with cardiorenal disease (CRD; blue) and without CRD (dotted blue) and in men (A) and women (B) with influenza between the years 2015 and 2019 with CRD (green) and without CRD (dotted green).

COVID-19 and CRD (53% vs 44% in women). Corresponding figures for influenza and CRD were 57% for men vs 49% in women.

When analysing the cumulative incidence of all-cause death in men (figure 3A) and women (figure 3B), there was no major difference. In addition, predictors for a fatal outcome with COVID-19 (illustrated in a forest plot, figure 4) were similar in both sexes with the strongest

association for age (10-year increase, 1.66; 95% CI 1.62 to 1.70, p<0.001 for men vs 1.55; 95% CI 1.51 to 1.60, p<0.001 for women), CRD (1.37; 95% CI 1.31 to 1.44, p<0.001 in men vs 1.46; 95% CI 1.38 to 1.54, p<0.001 in women), PAD (1.23; 95% CI 1.15 to 1.30, p<0.001 in men vs 1.30; 95% CI 1.21 to 1.40, p<0.001 in women), diabetes (1.10; 95% CI 1.05 to 1.15, p<0.001 in men vs 1.18; 95% CI 1.11 to 1.24, p<0.001 in women) and previous pneumonia

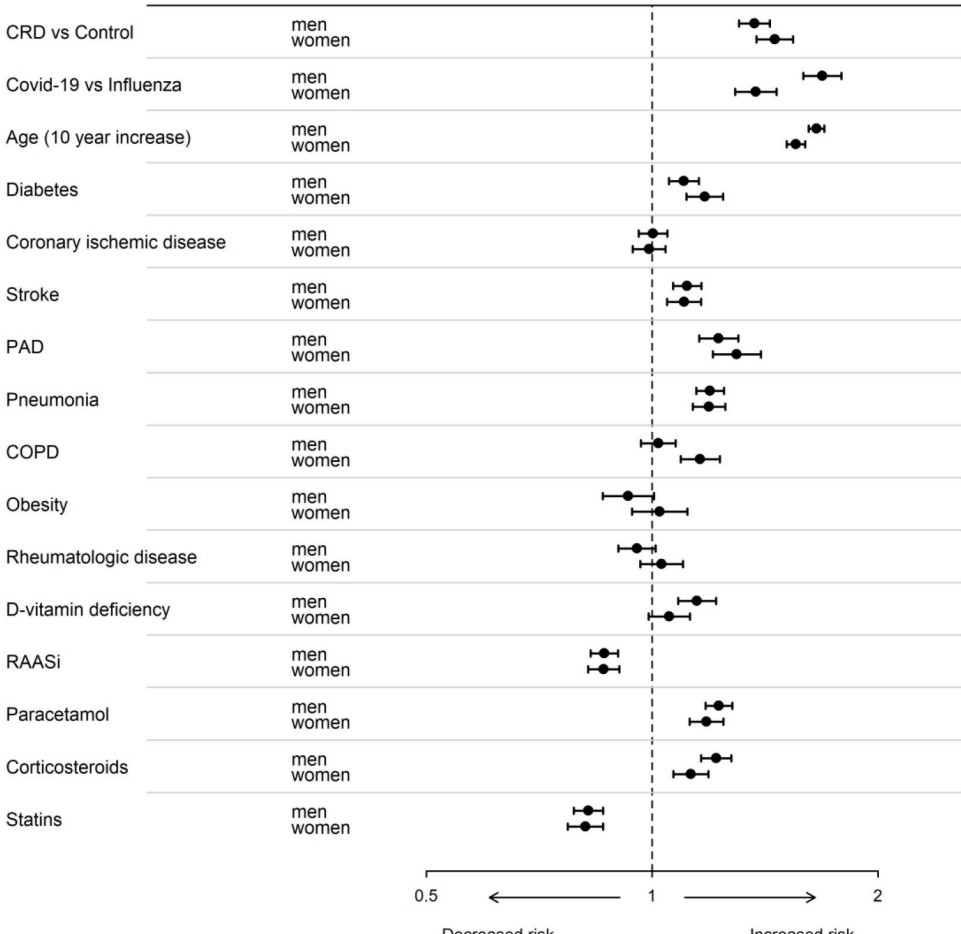

**Figure 4** Adjusted associated HR (95% CI) for all-cause death in men and women and mortality associated with COVID-19 versus influenza. COPD, chronic obstructive pulmonary disease; CRD, cardiorenal disease; PAD, peripheral artery disease; RAASi, renin–angiotensin system inhibitors.

(1.19; 95% CI 1.14 to 1.25, p<0.001 in men vs 1.19; 95% CI 1.13 to 1.25, p<0.001 in women). Previous treatment for hypertension and coronary artery disease, such as statins and RAAS inhibitors, were associated with reduced risk for all-cause death in both sexes.

### Impact of age and sex

When analysing all-cause death rate in men and women in specified age groups (<60 years, 60–69 years, 70–79 years, ≥80 years) CRD versus no CRD was associated with a consistent increased mortality rate after COVID-19 independent of sex and age (figure 5). Men had higher mortality rate than women regardless of CRD, apart from at ages <70 years, where men and women with CRD had similar mortality rate. In COVID-19 references without CRD, women had a decreased mortality rate compared with men at all age groups.

### DISCUSSION

In this nationwide study including all patients with CRD hospitalised for COVID-19 during all three waves in Sweden, there are several important findings.

First, hospitalisation for COVID-19 was associated with a 50% increased risk for mortality both in the CRD and the reference group compared with being hospitalised for influenza in the previous years illustrating that COVID-19 during the first waves was a more severe disease than influenza. The strongest predictor for mortality apart from age was CRD almost increasing the mortality risk by ~40% in patients with COVID-19. A history of CRD had a similar impact on patients hospitalised with influenza. We also identified PAD, previous hospitalisation for pneumonia and diabetes as important risk factors for a fatal outcome after COVID-19.

Second, overall female sex compared with male was protective from a fatal outcome following hospitalisation for COVID-19. However, in the presence of CRD this sex difference was abolished with an equal mortality rate in ages below 70 years.

This indicates that COVID-19 preventive measures should be prioritised in the elderly but also regardless of sex in those with CRD. Furthermore, our data strengthen that national strategies to prevent development of severe CRD could be of importance not only for preventing healthcare burden from cardiovascular complications but

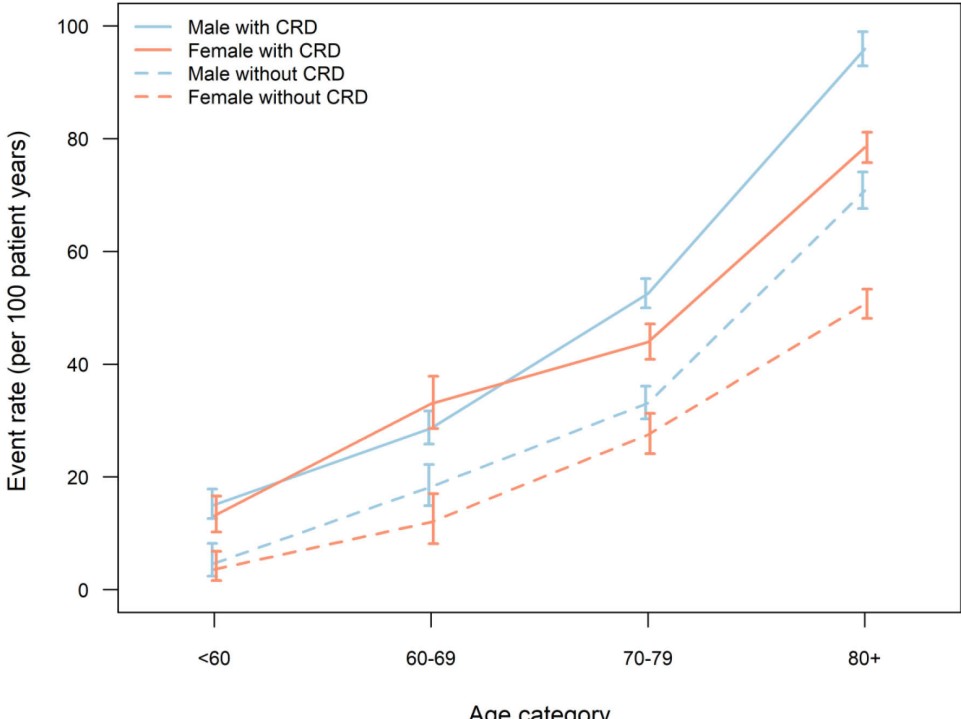

**Figure 5** All-cause death rate by sex and age groups stratified by cardiorenal disease (CRD) in patients with COVID-19.

also for reducing complications after severe infectious diseases as COVID-19 and influenza.

In Sweden, a high-income country with an older population and well-developed access to primary and secondary prevention, heart failure is present in 1%–2% and CKD in around 4%–6%.[14 15] In the Swedish population with age >70 years heart failure or CKD is present in around 10%–20%.[16] The CRD population (heart failure and/or CKD diagnosis prior to index) was compared with the reference group burdened with additional comorbidities as coronary artery disease present in 49% and diabetes 34% but when adjusting for other comorbidities CRD was independently associated with mortality.

Furthermore, a vast majority of the patients were treated with RAAS inhibitors and statins and thus considered to be on extensive pharmacological cardiovascular protection.

Hospitalisation for COVID-19 was associated with a significantly higher mortality risk during the first year after hospitalisation than that for hospitalisation for influenza during the previous years, especially during the first two waves where few had access to COVID-19 vaccine. This expands further knowledge from France on COVID-19 by Piroth *et al* showing an increased risk of in-hospital mortality compared with that of influenza.[17] The lower mortality rate in the third wave may visualise the effect of COVID-19 vaccination in vulnerable patients and improved experiences on optimal medical strategies. Vaccination is of vast importance to reduce mortality due to influenza. Of note, influenza vaccination in the elderly in Sweden is around 50% where those with CRD and

diabetes are prioritised and might have an even higher proportion of vaccination.[18]

As the ICD-10 code ZV100 (including procedures related to COVID-19) only has been used in Sweden, in particularly during the initial waves of the pandemic for COVID-19 and then more for procedures related to COVID-19, a sensitivity analysis was performed to enable comparison with international studies. Although the number of patients was reduced, the results were robust with an even higher death rate in COVID-19 patients regardless of CRD and with a greater associated mortality risk compared with that of influenza.

In the present cohort, CRD was an independent predictor for mortality while in contrast coronary artery disease was not. One explanation for this might be the extensive use of cardiovascular preventive pharmacological therapy. Further, the last decades, there has been a more successful prevention of incidence and complications from coronary artery disease, while prevention of incidence and complications from CRD has been less effective.[19–22] In addition, compared with an ischaemic event, CRD may have a diffuse disease pattern, and therefore, being clinically diagnosed later and in a more severe state. Previous treatments for hypertension and coronary artery disease, such as statins and RAAS inhibitors, were associated with a reduced risk for all-cause death in both sexes suggesting that well-controlled risk factors can prevent fatal outcome after COVID-19 infection. Accordingly, one could speculate that if CRD treatment could be improved and CRD being in a less severe state, outcome after a severe infection such as COVID-19 and also influenza would likely be less severe.

Unlike coronary artery disease, history of stroke and PAD were strong risk factors for a worse outcome probably indicating an underlying general severe atherosclerotic burden and frailty. Few studies have analysed the outcome of COVID-19 in patients with heart failure and/or CKD. In accordance with our results, Bergman et al[8] also found that in Sweden high age, diabetes, cardiovascular disease (however not analysing heart failure specifically), hypertension, renal failure and previous pneumonia were associated with all-cause death in hospitalised and non-hospitalised COVID-19 cases although with a shorter median follow-up time than ours (4 months).

Interestingly, we found that history of hospitalisation for pneumonia was common among COVID-19 patients, in particular, those with CRD (present in 39%), which was higher than previously reported by Bergman et al[8] (15%). Also, in the influenza population with CRD previous pneumonia was common (47%). After adjusting for age and other risk factors this was also one of the most important predictors for a fatal outcome. This does not seem to be explained by the presence of COPD which was less common (present in around one-fifth) and also a less important risk factor. Rather this seems to indicate a patient who is predisposed to a more severe disease development after a pulmonary infection. Our data are in line with a recent report from Ballin et al who analysed 30-day mortality in long-term care facility residents in Sweden and also identified previous pneumonia as an important risk factor for mortality in those diagnosed with COVID-19.[23]

Another Swedish study has reported on the selected group admitted to ICU and found diabetes, hypertension and obesity as independently associated risk factors for severe COVID-19, when having access to hospital recorded variables in the Swedish Intensive Care Registry.[7] In this study, obesity diagnosis (reported in the National Patient Register) was not a predictor for mortality which may reflect 'the obesity paradox' where overweight in heart failure and in CKD may indicate a less catabolic patient.[24] However, this should be interpreted with caution as we have no information about body mass index (BMI) and the reporting of obesity in the National Patient Register is of uncertain representativeness.

In patients with CRD hospitalised for COVID-19, almost half of the patients were women. Although significantly older, female sex was protective in survival analysis. However, in the presence of CRD this sex difference was abolished. At ages below 70 years, men and women had similar mortality rates but at ages above 70 men had a higher mortality rate than women. This pattern was not seen in the reference group where women had a lower mortality rate than men at all ages. Thus, in the presence of CRD, the protective effect of female sex was not present. Indeed, within women hospitalised for COVID-19 a history of CRD was identified as being the strongest risk factor for a fatal outcome. The presence of coronary artery disease, although less common in women, and previous hospitalisation for pneumonia were other risk factors for a fatal outcome in women with COVID-19.

The major strength of this study is the use of national registries enabling inclusion of all patients with CRD hospitalised for COVID-19 during all three waves in Sweden. Furthermore, we had the ability to compare COVID-19 with characteristics and outcome from previous influenza hospitalisations showing a higher mortality rate from COVID-19 when vaccination coverage still was low. Another strength is the comparison with a reference group with similar age and sex structure apart from CRD. A limitation is the inclusion criteria was restricted to hospitalised COVID-19 patients where patients that died with COVID-19 outside hospital care, as in elderly home care, have been missed. This is also true for the influenza cohort. Further, we have no individual data on influenza vaccination or proportion of patients vaccinated for influenza. Another limitation is the lack of information on laboratory analysis or variables as BMI as these registries do not contain such information. For example, obesity diagnoses are captured only from the National Patient Register, and therefore, associations between BMI and mortality could not be analysed in this study. However, in real-life situations, risk factor from medical history is often the only risk evaluation when prioritising vaccination and seldom based on a haemoglobin A1C (HbA1c) test or creatinine levels.

## Conclusion

We demonstrate that outcome in patients with COVID-19 and influenza is worse if prior CRD is present independently of age and sex. COVID-19 was associated with higher mortality risk than influenza during the first two waves. Furthermore, the presence of CRD abolished the protective effect of female sex in women at ages below 70 years. This study highlights the importance of preventing heart failure and CKD in the general population to reduce mortality from severe infections in the future.

**Author affiliations**
[1]Cardiology Unit, Department of Medicine, Solna, Karolinska Institute, Stockholm, Sweden
[2]Department of Research and Development, Region Kronoberg, Vaxjo, Sweden
[3]Cardiovascular, Renal and Metabolism, Medical Department, BioPharmaceuticals, AstraZeneca Nordic, Oslo, Norway
[4]Department of Medical Sciences, Clinical Diabetes and Metabolism, Uppsala University, Uppsala, Sweden
[5]Statisticon AB, Uppsala, Sweden
[6]Department of Medicine, Huddinge, Karolinska Institute, Stockholm, Sweden
[7]Department of Clinical Science and Education, Karolinska Institute, Södersjukhuset, Stockholm, Sweden
[8]Capio S:t Göran Hospital, Stockholm, Sweden

**Acknowledgements** The authors are grateful to Susanna Jerström and Helena Goike at AstraZeneca for logistic support and valuable comments on the manuscript. Urban Olsson, Statisticon AB, is acknowledged for database management.

**Contributors** The study design was developed by JB and MT. MT performed the statistical analyses and managed the database. VR and AN finalised the manuscript after adjustments by JB, MT, JE, RK, TN and DN. All authors made a substantial contribution to this article and were involved in the interpretation of the results. JB is the guarantor of this work.

**Funding** The study was fully sponsored by AstraZeneca. VR has received funding from the Department of Research and Development Region Kronoberg and the Family Kamprad Foundation. AN has received funding from the Family Kamprad Foundation and from Capio S:t Görans hospital.

**Competing interests** The authors declared the following potential conflicts of interest with respect to the research, authorship and publication of this article: VR has received honoraria on expert group participation from Astra Zeneca, Novo Nordisk and Boehringer Ingelheim. AN has received honoraria from Astra Zeneca, Merck Sharp & Dohme, Eli Lilly and Company, Novo Nordisk and Boehringer Ingelheim on expert group participation. TN has received unrestricted grants from AstraZeneca and NovoNordisk and has served on national advisory boards of Abbot, Amgen, Novo Nordisk, Sanofi-Aventis, Eli Lilly, MSD and Boehringer Ingelheim. JE has received research support or honoraria from AstraZeneca, NovoNordisk, Bayer, Ilya Pharma, Merck-Sharp & Dohme, Boehringer-Ingelheim. RK and DN reports no conflicts of interest. JB holds a full-time position at AstraZeneca as an epidemiologist. MT holds a full-time position by an independent statistical consultant company, Statisticon AB, Uppsala, Sweden, of which AstraZeneca Nordic is a client.

**Patient and public involvement** Patients and/or the public were not involved in the design, or conduct, or reporting, or dissemination plans of this research.

**Patient consent for publication** Not applicable.

**Ethics approval** The study was approved by the Stockholm Regional Ethics Committee (reference numbers 2020-05714 and 2013/2206-31). Ethical permission approved a reference group to a collected population with cardiorenal disease (CRD; heart failure and/or chronic kidney disease (CARE-19)) but not a reference group to COVID-19 or influenza patients.

**Provenance and peer review** Not commissioned; externally peer reviewed.

**Data availability statement** No data are available.

**ORCID iD**
Viveca Ritsinger http://orcid.org/0000-0003-3385-4777

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
