## [Reviewer comments · BMJ Open]

This paper was submitted to a another journal from BMJ but declined for publication following peer review. The authors addressed the reviewers' comments and submitted the revised paper to BMJ Open. The paper was subsequently accepted for publication at BMJ Open.

(This paper received two reviews from its previous journal but only one reviewer agreed to published their review.)

ARTICLE DETAILS

TITLE (PROVISIONAL)	History of heart failure and chronic kidney disease and risk of all-cause death after COVID-19 during the first three waves of the pandemic in comparison with influenza outbreaks in Sweden: a registry-based, retrospective, case-control study
AUTHORS	Ritsinger, Viveca; Bodegård, Johan; Kristofi, Robin; Thuresson, Marcus; Nathanson, David; Nyström, Thomas; Eriksson, Jan; Norhammar, Anna

VERSION 1 – REVIEW

REVIEWER	Reviewer 1
REVIEW RETURNED	N/A

GENERAL COMMENTS	Comment 1: This is an interesting study aimed to explore how cardiorenal disease (CRD; heart failure and/or chronic kidney disease) impacted mortality in patients hospitalised for COVID-19 during the three first waves in Sweden in comparison to previous influenza outbreaks and with a sex perspective. Comment 2: The Introduction lacks of the most recent literature on the association between COVID-19 and CKD (e.g. Oristrell J, Oliva JC, Subirana I, Casado E, Domínguez D, Toloba A, Aguilera P, Esplugues J, Fafián P, Grau M. Association of Calcitriol Supplementation with Reduced COVID-19 Mortality in Patients with Chronic Kidney Disease: A Population-Based Study. Biomedicines. 2021 May 5;9(5):509. doi: 10.3390/biomedicines9050509. PMID: 34063015; PMCID: PMC8147982). Please include. Comment 3: Table 1 should be stratified by sex because this is one of the main objectives of the study (i.e. with a sex perspective).
--

REVIEWER	Dr. Luiz Passaglia Clinical Hospital of UFMG
REVIEW RETURNED	N/A

GENERAL COMMENTS	Comment 1: The article has written according to Strobe standards, and with an appropriate statistical methodology. However, it loses originality in face of previously published information about this topic.
--

VERSION 1 – AUTHOR RESPONSE

Reviewer Comments: Reviewer: 1

Comments to the Author

Comment 1: This is an interesting study aimed to explore how cardiorenal disease (CRD; heart failure and/or chronic kidney disease) impacted mortality in patients hospitalised for COVID-19 during the three first waves in Sweden in comparison to previous influenza outbreaks and with a sex perspective.

Reply: We are thankful for this comment.

Comment 2: The Introduction lacks of the most recent literature on the association between COVID-19 and CKD (e.g. Oristrell J, Oliva JC, Subirana I, Casado E, Domínguez D, Toloba A, Aguilera P, Esplugues J, Fafián P, Grau M. Association of Calcitriol Supplementation with Reduced COVID-19 Mortality in Patients with Chronic Kidney Disease: A Population-Based Study. *Biomedicines*. 2021 May 5;9(5):509. doi: 10.3390/biomedicines9050509. PMID: 34063015; PMCID: PMC8147982). Please include.

Reply: We appreciate this suggestion, however the rational for this study was to explore how the prevalence of cardiorenal disease (heart failure and chronic kidney disease) impacts mortality after COVID-19 hospitalisation. The focus of the suggested article is on the association of treatment of CKD (i.e. calcitriol supplementation) on mortality. We have not included any intervention studies in the introduction and therefore refrain from adding the proposed study.

Comment 3: Table 1 should be stratified by sex because this is one of the main objectives of the study (i.e. with a sex perspective).

Reply: We agree on the importance of disclosing baseline characteristics for men and women, for clarity and to limit table size, we would keep them as two separate tables (found in Supplemental material)

Reviewer: 2

Comments to the Author

Comment 1: The article has written according to Strobe standards, and with an appropriate statistical methodology. However, it loses originality in face of previously published information about this topic.

Reply: We appreciate the acknowledgement of an appropriate statistical methodology. However we still believe that this study include several important findings and to the best of our knowledge we are not aware of any study making the comparison of mortality between COVID-19 and influenza in the different COVID-19 waves and in men and women with cardiorenal disease.